# Low-to-Moderate-Intensity Resistance Exercise Is More Effective than High-Intensity at Improving Endothelial Function in Adults: A Systematic Review and Meta-Analysis

**DOI:** 10.3390/ijerph18136723

**Published:** 2021-06-22

**Authors:** Yong Zhang, Ya-Jun Zhang, Hong-Wei Zhang, Wei-Bing Ye, Mallikarjuna Korivi

**Affiliations:** 1Department of Rehabilitation Medicine, Shaoxing University, Shaoxing 312000, China; 2021000025@usx.edu.cn; 2College of Physical Education and Health Sciences, Zhejiang Normal University, Jinhua 321004, China; ywbls@zjnu.cn (W.-B.Y.); mallik.k5@gmail.com (M.K.)

**Keywords:** endothelial function, resistance exercise, flow-mediated dilatation, meta-analysis

## Abstract

Aerobic exercise has been confirmed to improve endothelial function (EF). However, the effect of resistance exercise (RE) on EF remains controversial. We conducted this systematic review and meta-analysis on randomized controlled trials (RCTs) to determine the effect of RE and its intensities on EF. We searched Web of Science, PubMed/MEDLINE, Scopus, and Wiley Online Library, and included 15 articles (17 trials) for the synthesis. Overall, RE intervention significantly improved flow-mediated dilatation (FMD) in brachial artery (SMD = 0.76; 95% CI: 0.47, 1.05; *p* < 0.00001), which represents improved EF. Meta-regression showed that the RE intensity was correlated with changes in FMD (Coef. = −0.274, T = −2.18, *p* = 0.045). We found both intensities of RE improved FMD, but the effect size for the low- to moderate-intensity (30–70%1RM) was bigger (SMD = 1.02; 95% CI: 0.60, 1.43; *p* < 0.0001) than for the high-intensity (≥70%1RM; SMD = 0.48; 95% CI: 0.21, 0.74; *p* = 0.005). We further noticed that RE had a beneficial effect (SMD = 0.61; 95% CI: 0.13, 1.09; *p* = 0.01) on the brachial artery baseline diameter at rest (BAD_rest_), and the age variable was correlated with the changes in BAD_rest_ after RE (Coef. = −0.032, T = −2.33, *p* = 0.038). Young individuals (<40 years) presented with a bigger effect size for BAD_rest_ (SMD = 1.23; 95% CI: 0.30, 2.15; *p* = 0.009), while middle-aged to elderly (≥40 years) were not responsive to RE (SMD = 0.07; 95% CI: −0.28, 0.42; *p* = 0.70). Based on our findings, we conclude that RE intervention can improve the EF, and low- to moderate-intensity is more effective than high-intensity.

## 1. Introduction

Vascular endothelial dysfunction is an independent risk factor for predicting cardiovascular diseases (CVDs), and is closely associated with the occurrence of several CVDs, such as hypertension, atherosclerosis, heart failure (stroke), and metabolic syndrome [1,2,3]. Impaired endothelial function (EF) can lead to the deterioration of the blood vessel wall, and eventually accelerate the atherosclerotic process, and is used as an overall physiological indicator of vascular health [4]. Flow-mediated dilatation (FMD) is the most commonly used non-invasive clinical method to determine EF. It uses the ultrasound method to measure the diameter of the arteries at baseline, as well as after reactive hyperaemia secondary to temporary occlusion, and calculates the percentage increase of the peak diameter compared with baseline. The reliability and validity of the brachial artery FMD has been confirmed as a valid marker for assessing cardiovascular risk [5,6], and the data support the stability and reproducibility of brachial FMD [6]. A systematic review and meta-analysis of 17,280 participants showed that a 1% increase in FMD can reduce the risk of cardiovascular events by 12% [7]. Therefore, improving EF with exercise training is an important strategy to decrease cardiovascular complications and promote health.

Aerobic exercise training is an effective evidence-based strategy to reduce CVD risk factors, possibly by promoting EF and arterial stiffness. Several studies have confirmed the efficacy of aerobic exercise in improving the EF in healthy young adults [8,9,10], older adults [11,12,13], and unhealthy men [14,15,16,17,18,19,20,21]. It is further emphasized that the greater beneficial effects of exercise intervention on EF are associated with exercise modalities and intensity [13,21,22,23]. However, the effect of resistance exercise (RE) on EF is yet to be elucidated. Some studies have reported that RE can improve EF [16,19,24,25,26,27,28,29,30], while other studies claimed that RE had no effect on EF [13,17,20,31,32,33,34,35]. In contrast, Otsuki et al. reported increased arterial stiffness and elevated endothelin-1 levels in strength trained men, which indicates impaired EF [36]. A systematic review and meta-analysis concluded that the RE frequency was positively associated with improved EF rather than intensity [22]. Despite these inconsistent findings, we speculate that RE variables, as well as the age and health status of participants, may contribute to the differences in EF. 

To the best of our knowledge, there is no systematic review and meta-analysis of randomized controlled trials (RCTs) to explore the effect of RE intensity on EF in adults. Therefore, we designed this systematic review and meta-analysis to investigate the influence of RE and its characteristics on improving the EF. We used brachial artery FMD as an outcome to determine whether the RE intensity improved the EF in adults. In addition, the effective intensity of RE was identified through subgroup analysis.

## 2. Methods

### 2.1. Searching Processes

We followed the Preferred Reporting Items for Systematic Reviews and Meta-Analysis (PRISMA) guidelines [37] to conduct this systematic review and meta-analysis. 

The article search was conducted through electronic databases, such as Web of Science, PubMed, MEDLINE, Scopus, and Wiley Online Library. Articles published until 8 January 2021 were searched for using the following keywords: “resistance” and “endothelial function”, and “exercise”/“training”/“physical activity” in the title and abstract. The main keywords of “exercise”, “training”, or “physical activity” were independently used with “resistance”, “endothelial function”, “flow-mediated dilatation”, “FMD”, and “shear stress”, and three searches were performed separately. During the search process, we applied the filtering function of the database using “article”, “randomized controlled trial”, and “journal” options wherever applicable. 

### 2.2. Article Inclusion and Exclusion Criteria 

Two authors (Y.Z. and Y.-J.Z.) independently performed the article search, article selection, data extraction, and assessment of the quality. Another two authors, H.-W.Z. and W.-B.Y., provided additional review and insight. Any disagreement in article inclusion and exclusion were discussed with the other author (M.K.) and were resolved. 

Initially, the titles and abstracts of the identified articles were thoroughly screened for relevance, and then the full-text of the selected articles was procured and reviewed. The following criteria were used to include the articles in this review: (1) studies were RCTs and published in English; (2) resistance exercise training was the only intervention in the trials, not combined with any other intervention; (3) the control trials did not participate in any exercise, and maintained daily behavior or were sedentary; (4) RE duration was 4 weeks or more; and (5) the outcome of assessing EF was brachial artery FMD in adults. The exclusion criteria were as follows: (1) non-RCTs or without a control group; (2) non-resistance training or combined with other interventions; (3) animal studies; (4) acute RE studies; (5) studies without FMD data; and (6) repeated research reports or data, non-English, inadequate details of RE, or low quality (articles in non-peer reviewed journals or preliminary reports). The detailed process of the article selection and inclusion criteria is schematically presented in Figure 1. 

### 2.3. Extraction of Data from the Included Studies

Information and data from the 15 eligible articles (17 trails) were independently extracted by three of the review authors (Y.Z., Y.-J.Z., and M.K.). The data, including the basic information of the study (authors, publishing year, and study performed country), characteristics of the subjects (age, sex, and health status), variables of RE (intensity, repetitions, sets, frequency, and duration), and outcomes are presented in Table 1. The outcome values were presented as mean and standard deviation (SD). If the mean and SD data were not available in the article, we contacted the corresponding author for further information. If no response was received from the author, standard errors (SE) were converted to SD, and the quartile data were converted to mean and SD [38,39]. The data reported in the table were extracted to the nearest numbers using the WebPlotDigitizer (https://automeris.io/WebPlotDigitizer/).

### 2.4. Assessment of Risk of Bias 

In this study, we used the Cochrane Collaboration tool to assess the risk of bias [40]. For the assessment, full-text information of the included articles was carefully reviewed and we applied the risk of bias tool. Initial assessment was done by two authors (Y.Z. and Y.-J.Z.), and any disagreements were rectified by discussing with other author (M.K.). The source of bias, including selection (allocation concealment and random sequence generation), performance (blinding of participants and personnel from exercise intervention), detection (blinding of outcome assessment), attrition (incomplete outcome data), reporting (selective reporting), and other biases were detected. The results of the risk of bias are explained in the results section.
ijerph-18-06723-t001_Table 1Table 1Characteristics of the included studies.StudyCountryHealth StatusAge (Y)Participants (M/F)Description of REIntensity (%1RM)RepetitionsSetsFrequency (t/wk)Duration (wk)OutcomeRE/ControlREControlBeck et al., 2013 [24]AmericaYoung prehypertensive 21.1 ± 2.5/ 21.6 ± 2.915 (11/4)15 (10/5)Leg extension, leg curl, leg press, lat pull down, chest press, overhead press, and biceps curl50%8–12238FMD; ↑ BAD_rest_↔Boeno et al., 2020 [19]BrazilMiddle-aged hypertensive patients30–5915 (nr)12 (nr)Bench press, leg press, lat pulldown, leg extension, shoulder press, leg curl, bicep curl and triceps extension; abdominal crunches were performed during each session65%8–202–3312FMD; ↑ BAD_rest_↔Casey et al., 2007 [31]AmericaHealthy21 ± 2.45/ 22 ± 2.9724 (11/13)18 (8/10)Leg extension, leg curl, leg press, lat pulldown, chest press, overhead press, and bicep curl70%8–122312FMD↔Franklin et al., 2015 [25]AmericaObese women30.3 ± 5.4/ 30.8 ± 9.010 (0/10)8 (0/8)8–10 dynamic-resistance exercises in sequence with a 30s rest period between exercises, to target the major muscle groups of the upper and lower body65%8–102–328FMD; ↑ BAD_rest_↔Hildreth et al., 2018 [32] AmericaHealthy older men66 ± 5/ 67 ± 519 (19/0)21 (21/0)Four upper- and three lower-body exercises80%6–83348FMD; ↔ BAD_rest_↔Hildreth et al., 2018 [32]AmericaHealthy older men66 ± 5/ 67 ± 519 (19/0)21 (21/0)Four upper- and three lower-body exercises80%6–83324FMD; ↔ BAD_rest_↔Jaime et al., 2019 [26]AmericaPostmenopausal nonobese women64 ± 3.46/ 67 ± 2.8312 (0/12)8 (0/8)Leg press, leg extension, leg flexion, and calf raise40%152312FMD; ↑ BAD_rest_↔Kwon et al., 2011 [17]KoreaOverweight type 2 diabetes56.3 ± 6.1/ 58.9 ± 5.712 (0/12)15 (0/15)Upper body exercises: bicep curls, tricep extensions, upright rows, shoulder chest press, and seated rows. Core exercises: trunk side bends, and lower body exercises included a leg press, hip flexions, leg flexions, and leg extensions40–50%nr3312FMD↔McDermott et al., 2009 [33] AmericaPeople with peripheral arterial disease71.7 ± 8.7/ 68.5 ± 11.936 (nr)28 (nr)Knee extension, leg press, and leg curl exercises using standard equipment80%83324FMD↔Okamoto et al., 2008 [28]JapanHealthy young men19.4 ± 0.2/ 19.4 ± 0.210 (10/0)9 (9/0)Chest press, arm curl, lateral pull down, seated row, shoulder press, leg extension, leg curl, leg press and sit-up (3s lowering phase and 3s lifting phase)40%10528FMD; ↑ BAD_rest_↑Okamoto et al., 2009 [34]JapanHealthy young adults19.6 ± 1.26/ 19.7 ± 0.9510 (10/0)10 (10/0)Chest presses, arm curls, seated rowing, leg curls, leg presses and sit-ups (1s lifting phase and 3s lowering phase)80%8–105210FMD; ↔ BAD_rest_↑Okamoto et al., 2009 [34]Japanhealthy young adults19.2 ± 0.95/ 19.7 ± 0.9510 (10/0)10 (10/0)Chest presses, arm curls, seated rowing, leg curls, leg presses and sit-ups (3s lifting phase and 1s lowering phase)80%8–105210FMD; ↔ BAD_rest_↑Okamoto et al., 2011 [27]JapanHealthy young 18.5 ± 0.5/ 18.6 ± 0.513 (10/3)13 (9/4)Chest press, arm curl, seated row, lateral pull down, leg press, leg extension, leg curl, and sit-ups50%105210FMD; ↑ BAD_rest_↑Olson et al., 2006 [29]AmericaOverweight women38 ± 3.87/ 38 ± 7.7515 (0/15)15 (0/15)RE was performed using isotonic variable resistance machines and free weights targeting the following major muscle groups: quadriceps, hamstrings, gluteals, pectorals, latissimus dorsi, rhomboids, deltoids, biceps, and triceps.75%8–103248FMD; ↑ BAD_rest_↔Rech et al., 2019 [35]BrazilElderly individuals with T2DM70.5 ± 7.4/ 68 ± 6.517 (10/7)21 (10/11)Partial squat, bench stepping, unilateral leg press, unilateral knee extension, knee flexion, plantar flexion, bench press, low row, biceps curl, elbow extension, hip abduction and abdominal crunches.60%10–122–3312FMD; ↔ BAD_rest_↔Vona et al., 2009 [16]SwitzerlandPatients with recent myocardial infarction57 ± 8/ 58 ± 754 (39/15)50 (37/13)Chest press, shoulder press, triceps extension, biceps curl, pull-down [upper back], lowerback extension, abdominal crunch/curl-up, quadriceps extension or leg press, leg curls [hamstrings], and calf raise.60%10–12444FMD; ↑ BAD_rest_↔Yu et al., 2016 [30]ChinaHealthy adolescents12.3 ± 0.42/ 12.1 ± 0.3019 (12/7)19 (13/6)Elbow extension, elbow flexion, trunk extension, trunk flexion, shoulder press, knee extension, knee flexion, push ups, squats, incline dip, hip abduction, and hip adduction70%124210FMD↑Y, years; RE, resistance exercise; C, control; M/F, male/female; 1RM, one-repetition maximum; t/wk, times/week; FMD, flow-mediated dilation; BAD_rest_, brachial artery baseline diameter at rest; ↑, increased; ↔, not changed; nr, not reported.

### 2.5. Outcomes and Subgroups Division

As a common clinical approach for assessing EF, the brachial artery FMD data were included for the meta-analysis. FMD is expressed as the percentage of changes in the arterial diameter from resting baseline to the post-ischemic peak (FMD%). Both the brachial artery diameter and FMD may have similar predictive values for the incidence of cardiovascular events in older adults [41]. The brachial artery baseline diameter at rest (BAD_rest_) was also included in this meta-analysis.

Meta-regression analysis was performed to identify the association between exercise variables and outcomes. We found the change in brachial artery FMD after RE was correlated with the RE intensity variable (Coef. = −0.274, T = −2.18, *p* = 0.045), and not with sets, frequency, duration, age, and health status. Then, to identify the effective RE intensity, we categorized the trials into high-intensity (≥70% 1RM) and low- to moderate-intensity (30–70% 1RM) subgroups [42]. In addition, changes in BAD_rest_ after RE intervention were correlated with the age of the participants (Coef. = −0.032, T = −2.33, *p* = 0.038), and not were correlated with the RE variables (intensity, sets, frequency, and duration) and health status of the participants. We therefore categorized the trials into young (<40 years old) and middle-aged to elderly (≥40 years old) [43] to identify the influence of age on BAD_rest_ after RE intervention.

### 2.6. Statistical Analyses

The Cochrane Collaboration Review Manager (RevMan, Copenhagen, Denmark) version 5.3 was employed for the statistical analysis. The primary statistical procedures in this study were the computation, heterogeneity, and verification of combined effect size. If no significant difference (*p* > 0.05) was noticed in the heterogeneity analysis, the fixed effect model was adopted for the meta-analysis. Incase heterogeneity was significant (*p* < 0.05), the random effect model was used for the analysis. The STATA version 12.0 (Stata Corporation, College Station, TX, USA) was used for the sensitivity analyses, publication bias, and meta-regression analysis. 

In this study, the standardized mean difference (SMD) was used to decide the magnitude of the intervention effect on the outcomes. Where the change was <0.2, it was defined as trivial, 0.2 to 0.3 was defined as small, 0.4 to 0.8 was defined as moderate, and >0.8 was defined as a large effect [44]. The SMD was expressed as a 95% confidence interval (CI). The statistical heterogeneity across the included trials in the meta-analysis was determined by the I^2^ statistic, where <25% implies a low risk of heterogeneity, 25 to 75% indicates a moderate risk of heterogeneity, and >75% denotes a considerable risk of heterogeneity [45]. 

## 3. Results

### 3.1. Summary of the Included Articles

A total of 813 related articles were identified from the electronic databases, including Web of Science, PubMed, MEDLINE, Scopus, and Wiley Online Library. After screening the titles and abstracts, 758 articles were excluded and 55 were selected for the full-text assessment. Finally, 15 articles, which fulfilled the required inclusion criteria, were included in this study. The article search, exclusion, and selection in each step are summarized in Figure 1.

The included articles (n = 15) were published between 2006 and 2020. Of these, seven studies were conducted in the USA [24,25,26,29,31,32,33], three were from Japan [27,28,34], two were from Brazil [19,35], one was from Korea [17], one was from Switzerland [16], and one was from China [30]. Regarding gender participation, three studies recruited only males, four studies recruited only females, six studies enrolled a combination of both, and two studies did not report this. The number of participants in the RE trials ranged from 10 to 54, and the number of participants in the control trials ranged from 8 to 50. The total number of participants (sample size) in the RE and control trials were 310 and 293, respectively. Detailed characteristics of the RE intervention and participants are shown in Table 1.

### 3.2. Risk of Bias of the Included Studies

The detailed judgment of risk of bias derived from the Cochrane Collaboration tool is depicted in Figure 2. For the included 17 trials, we found a low risk of selection bias (random sequence generation and allocation concealment). Sixteen trials reported to have a high risk of performance bias, and one study conducted a double-blind design in exercise intervention [35] and was judged to have an unclear risk of bias. Blinding participants in an exercise intervention is usually difficult. Therefore, reporting a high risk of performance bias does not compromise the quality of the included studies [46,47]. However, other key variables, such as the study attrition level, poor adherence to intervention, and selective reporting bias, commonly take place when there is a high risk of bias, and possibly influence the quality of the study [48]. No study was judged to have a high risk of detection bias, attrition bias, reporting bias, or other bias in our assessment.

We used STATA for the sensitivity analyses and publication bias analyses. After omitting one study, the estimates were all within the combined confidence interval (Estimate = 0.757, 95% CI = 0.47–1.05). No trial was found to have a significant impact on the total effect size of the FMD. The funnel plot and Egger linear regression test (Egger’s test, t = −0.82; *p* = 0.425; 95% CI = −5.23, 2.33) also showed no publication bias in the RE interventions for FMD. Similarly, the sensitivity analysis did not find any significant impact on the total effect size of BAD_rest_ (combined estimate = 0.757, 95% CI = 0.47–1.05). However, the funnel plot and Egger linear regression test (Egger’s test, t = 2.60; *p* = 0.023; 95% CI: 0.67, 7.63) showed a publication bias. From the funnel plot, it is clear that one trail had a relatively large bias [24], which reported significantly increased BAD_rest_ after RE intervention (SMD = 4.58, 95% CI: 3.15, 6.01).

### 3.3. Low- To Moderate-Intensity RE Is Effective than High-Intensity RE in Improving FMD

In 17 trials, brachial artery FMD was used to assess the EF. We used SMD to determine the effect size of RE on FMD change under a random effect model. We found that RE had a significant effect on FMD improvement (SMD = 0.76; 95% CI: 0.47, 1.05; *p* < 0.00001). However, a moderate heterogeneity was noticed in the studies (I^2^ = 62%, *p* = 0.0004).

We then performed a subgroup analysis and found significantly improved FMD with both intensities of RE (low- to moderate-intensity and high-intensity). In particular, the effect size with a low- to moderate-intensity was bigger (SMD = 1.02; 95% CI: 0.60, 1.43; *p* < 0.0001), and the risk of heterogeneity was moderate (I^2^ = 61%, *p* = 0.009). Importantly, the effect size in each study was above the medium level. A high-intensity RE showed a moderate effect on FMD improvement (SMD = 0.48; 95% CI: 0.21, 0.74; *p* = 0.005). Here, we noticed that the effect size of low- to moderate-intensity was relatively bigger than the effect size of high-intensity. Furthermore, the test for subgroup differences (in the effect of size of RE on FMD change) between low- to moderate-intensity and high-intensity were statistically significant (Chi^2^ = 4.64, I^2^ = 78.4%, *p* = 0.03). These findings indicate that improved FMD with a low- to moderate-intensity RE is more effective than high-intensity RE.

### 3.4. Resistance Exercise Improves BAD_rest_ in Young, but Not in Middle-Aged to Elderly

We used SMD to determine the effect size of RE on BAD_rest_ under a random effect model. The results showed that RE had a significant effect on improved BAD_rest_ (SMD = 0.61; 95% CI: 0.13, 1.09; *p* = 0.01), but there was a considerable high risk of heterogeneity in these studies (I^2^ = 81%, *p* < 0.00001). Considering the influence of age on BAD_rest_ changes (Coef. = −0.032, T = −2.33, *p* = 0.038), we conducted a subgroup analysis by categorizing the trials into young adult (<40 years) and middle-aged to elderly (≥40 years) subgroups [43]. The results showed a significant difference in the effect size of RE on BAD_rest_ change between young adults and middle-aged to elderly (Chi^2^ = 5.26, I^2^ = 81.0%, *p* = 0.02). RE had a bigger effect on BAD_rest_ of young adults (SMD = 1.23; 95% CI: 0.30, 2.15; *p* = 0.009), but there was a considerable risk of heterogeneity in these studies (I^2^ = 85%, *p* < 0.00001). In the middle-aged to elderly group, no significant effect of RE was found (SMD = 0.07; 95% CI: −0.28, 0.42; *p* = 0.70) (Figure 3).

## 4. Discussion

In our systematic review and meta-analysis, the brachial artery FMD, a typical clinical measure to evaluate EF, was used as an outcome in RE trials. The meta-analysis results revealed that RE intervention significantly improved EF in adults, as evidenced by the increased FMD, but the heterogeneity was moderate. We performed meta-regression analysis, and found that the “intensity” variable of RE is correlated with the changes of the brachial artery FMD. We then performed a subgroup analysis to identify the effective intensity of RE. Both low- to moderate-intensity and high-intensity were able to improve EF; however, the improvement was greater with low- to moderate-intensity than that of high-intensity. These results are different from the aerobic exercise interventions. In aerobic exercise trials, increased exercise intensity is reported to have greater beneficial effects on EF [13,21,22,23]. A meta-analysis also reported a significant dose-response relation between aerobic exercise intensity and FMD [22]. However, our findings revealed that low- to moderate-intensity RE is more effective than the high-intensity at improving EF in adults. 

As of now, no meta-analysis was conducted using RCTs with long-term RE intervention on EF. This is the first systematic review and meta-analysis to demonstrate the influence of RE and its intensities on FMD in adults. A cross-sectional designed study found that strength-training increased arterial stiffness and elevated endothelin-1 levels in trained men, which indicates impaired EF [36]. Few meta-analyses have attempted to delineate the role of physical activity or the combination of aerobic exercise and RE on endothelial functioning and arterial stiffness. Nevertheless, the available meta-analyses of RE intervention did not address the impact of RE and its intensities on EF or FMD changes in young and middle-aged adults. Qiu and team conducted a meta-analysis on RE intervention and EF of patients with type 2 diabetes. However, they included only one RE study in their analysis, and no statistical data were provided on the influence of intervention variables [18]. Another meta-analysis by Ashor et al. reported that RE can improve FMD in adults, but with a quite high heterogeneity (I^2^ = 91.6%, *p* < 0.001). Besides, two of the 12 included trials in their meta-analysis reported to have a negative effect size, and should be interpreted with caution [22]. It should also be noted that the effect size of RE in their analysis was not related to the intensity, but it was related to the frequency [22], which is different from our findings. Our study showed a greater improvement of EF with a low-to-moderate-intensity than that of high-intensity RE, which may aid in promoting cardiovascular health in adults.

The precise reason and mechanism behind the improvement of EF with RE intervention could be an interesting topic to be investigated. Vascular endothelium plays a vital role in the regulation of normal vascular tone through the release of vasoactive substances, such as nitric oxide (NO) and endothelin-1. NO is a labile, lipid soluble gas produced by endothelial nitric oxide synthase (eNOS) in endothelial cells from L-arginine. NO is released in response to several factors, including shear stress, and produces vascular relaxation through diffuses into smooth muscle cells [49,50]. The literature revealed that exercise-induced increased shear stress might be a responsible factor for the improvement of EF [51,52]. During an exercise session, in response to the high metabolic demand, there will be a large increase in blood-flow supply to the active muscles. This scenario may lead to an increase in the shear rate, and trigger the NO production and/or increase its bioavailability [53]. These findings indicate that increased exercise intensity is associated with increased shear stress, increased bioavailability or production of NO, and a beneficial effect of vasodilation. NO and endothelin-1, both produced by endothelial cells, are the responsible molecules to regulate vascular function. The physiological imbalance between NO and endothelin-1 molecules represents endothelial dysfunction [54]. Besides, a recent study demonstrated potassium (K^+^) ion as an important vasodilator agent contributing to exercise hyperemia in humans [55].

In fact, different exercise modalities are associated with different patterns of luminal shear stress, and differing shear patterns contribute to a diverse response of vascular function [51,56,57]. Some studies reported that increased antegrade shear is typically linked with improved vascular function and upregulated NO signaling [51,57,58]. In contrast, increased retrograde, and turbulent shear may decrease the NO bioavailability and the generation of a proatherogenic phenotype [59,60]. Tinken et al. reported that the magnitude of the increased antegrade shear rate was concomitantly correlated with improved FMD in healthy men [51]. Furthermore, Thijssen et al. [57] and Johnson et al. [61] reported that increased levels of retrograde shear may contribute to decreasing the FMD. We agree that RE training increases blood flow and antegrade shear with the low- and medium-intensity [51]. However, during high-intensity RE, muscle contraction is enhanced and relaxation is weakened, which significantly enhances retrograde and turbulent shear, and causes an adverse effect on the upregulation of the NO pathway.

In addition, the high-intensity RE is usually accompanied by a greater increase in blood pressure [62,63,64], which is detrimental to FMD by down-regulating the cNOS levels and up-regulating the endothelin-1 [64,65,66,67,68,69,70]. Such an occurrence with high-intensity RE affects the EF. Buchanan et al. [69] demonstrated that the attenuation of exercise-induced elevated blood pressure abolished the brachial artery endothelial dysfunction in sedentary individuals. These findings support the notion that high-intensity resistance training-induced endothelial dysfunction is actually mediated by elevated blood pressure. Morishima et al. [71] also confirmed that excessive elevation in blood pressure is not conducive to EF. Conversely, some studies have revealed that low-intensity resistance training is beneficial for controlling the blood pressure [17,24,72], which may explain the advantages of low- to moderate-intensity RE to improve the EF.

Other potential mechanisms include oxidative stress, sympathetic nerve activity, etc. Oxidative stress is an imbalance between pro-oxidants (free radicals) and antioxidants that favors pro-oxidant status over antioxidants. Chronic exposure of highly reactive free radicals eventually causes vascular dysfunction and/or damage. It has been documented that increased oxidative stress causes more NO to participate in free radical scavenging, which reduces the bioavailability of NO in vasodilation. Exercise intervention with higher intensities has been shown to trigger free radical production and cause severe oxidative stress [73], which is detrimental to the improvement of EF. Similarly, there is a competing mechanism between the endothelium-dependent dilator capacity and sympathetic constriction. When high-intensity exercise increased the sympathetic nerve activity, the adrenal medulla secretes noradrenaline that constricts blood vessels, and contributes to increased blood pressure [74]. Of course, long-term regular RE with a low- to moderate-intensity can improve the antioxidant capacity and reduce oxidative stress, and thereby promote the adaptation of EF and maintain the balance between sympathetic and parasympathetic activity.

As the baseline diameter is included in the FMD calculation, changes in the baseline diameter induced by exercise should be considered in the measurement of EF. A study recognized that the brachial artery diameter and FMD had similar predictive values for cardiovascular events [41]. Another study concluded that using only FMD% may lead to bias [75]. Therefore, BAD_rest_ was also analyzed as an outcome in our meta-analysis. From the results, we found that RE did not improve the BAD_rest_ of the middle-aged to elderly. However, young individuals presented with increased BAD_rest_, despite the risk of heterogeneity. The increase BAD_rest_ in young adults is considered to be an adaptation that may be mediated by persistent chemical stimuli, and these chemical stimuli affect arterial tone and/or physical structural changes to the vessel itself [76]. In addition to improved endothelial cell function (up-regulation of NO and down-regulation of endothelin-1), increased antioxidant capacity and improved sympathetic and parasympathetic balance may also contribute to this change in young adults. Generally, healthy aging is accompanied by an increase in basic sympathetic nerve activity. Moreover, the responsiveness of sympathetic nerve and parasympathetic nerve activity also decreases with aging [77]. These changes in sympathetic and parasympathetic nerves in the aging process may affect the increase of BAD_rest_ after RE in middle-aged and elderly people. A previous study reported endurance exercise (8-week) did not alter the characteristics of brachial or carotid arteries and FMD in sedentary older men [11]. A meta-analysis reported a greater baseline brachial artery diameter in young athletes (<40 years), but not in master athletes (>50 years) [78]. Older adults are typically represented with an impaired antioxidant status or increased oxidative stress and inflammation, which subsequently may contribute to impair the vascular function in older adults [79]. The deterioration of the vascular structure in the elderly may be a considerable reason for the unchanged BAD_rest_. Further research is needed to evaluate the longitudinal effects of RE on vasoactive substances, as well as vascular function and structural remodeling in middle-aged and elderly people.

## 5. Conclusions

The evidence from this systematic review and meta-analysis demonstrates that RE can improve endothelial function. Particularly, low- to moderate-intensity RE is more effective at improving EF than that of high-intensity. RE cannot improve the baseline diameter of the brachial artery at rest in middle-aged to elderly, and its improving effect on the resting diameter of the brachial artery in young adults is still quite heterogeneous, and further research is necessary.

## Figures and Tables

**Figure 1 ijerph-18-06723-f001:**
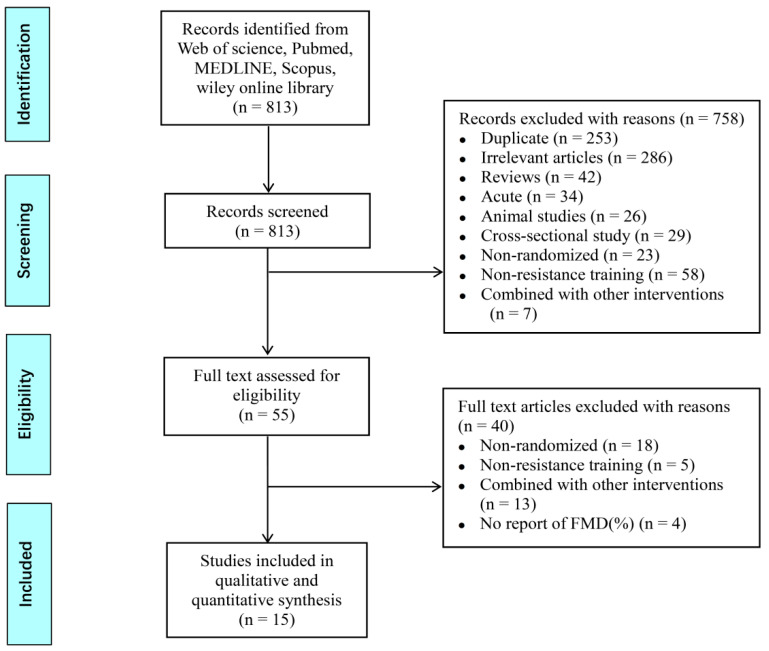
Flow-chart of study selection and exclusion according to the Preferred Reporting Items for Systematic Review and Meta-analysis (PRISMA).

**Figure 2 ijerph-18-06723-f002:**
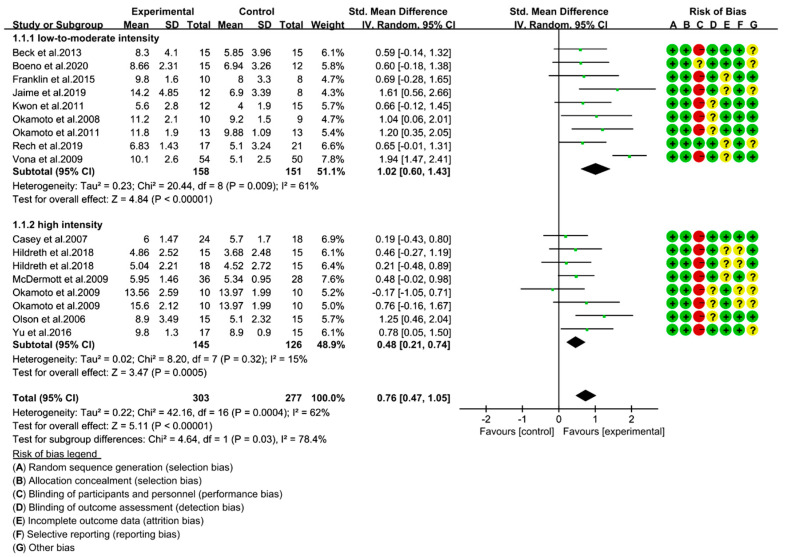
Forest plot of the brachial artery FMD change with different intensities of RE (%). Risk of bias; low risk of bias (+ green), high risk of bias (− red), and unclear risk (? yellow).

**Figure 3 ijerph-18-06723-f003:**
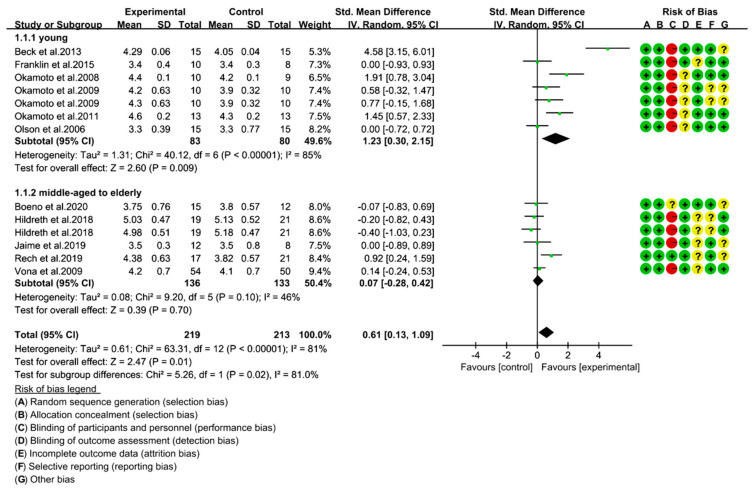
Forest plot of the brachial artery baseline diameter at rest (BAD_rest_) in different age groups (mm). Risk of bias; low risk of bias (+ green), high risk of bias (− red), and unclear risk (? yellow).

## Data Availability

The datasets generated and analyzed for this study can be requested from the correspondence authors at YajunZhang@usx.edu.cn (Y.-J.Z.) and zhanghw@usx.edu.cn (H.-W.Z.).

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
