# Peer review of "Low-to-Moderate-Intensity Resistance Exercise Is More Effective than High-Intensity at Improving Endothelial Function in Adults: A Systematic Review and Meta-Analysis"

_ijerph, 2021, doi:10.3390/ijerph18136723_

Round 1

Reviewer 1 Report

Well written article, congratulations to the authors. I believe it is suitable for publication in this important scientific journal. In the meantime, I would like to make some suggestions. In lines 73 and 74 I suggest using the terms "shear stress", "FMD" and "flow-mediated dilatation" associated with the terms "training", "exercise" and "physical activity" in the research to find more work on the topic.

 In line 94 I suggest explaining the concept of "low quality" so that the reader understands the criterion perfectly, since the term used leaves a vague idea of ​​what it would be. In lines 221 to 225, the authors could present a stratification of ages over 40 years. Between 40 years and 60 years for 20 years. After the fourth decade of life, 10 years generate very important changes, so I suggest that the authors present this data and discuss it in the discussion.

In lines 271 to 282 the authors present NO, however, they do not discuss the synthesis and release of this gas. I suggest that you proceed a little further by presenting the synthesis and liberation pathways. Finally, there are other mechanisms that influence endothelial function, such as potassium channels. See the works by Gree et al (2017) and Terwoord et al (2018), attached articles.

Author Response

Responses to Reviewers’ Comments: ijerph-1237253

Reviewer 1

Comment: Well written article, congratulations to the authors. I believe it is suitable for publication in this important scientific journal. In the meantime, I would like to make some suggestions.

Response: We are highly thankful to the Reviewer for encouraging words and valuable comments on our manuscript. We agree with the comments and all comments helped us to improve the quality of our manuscript. According to the comments, we have carefully revised our manuscript, and all corrections were marked in red color. Our point-by-point response to each comment was given below: 

Comment: In lines 73 and 74 I suggest using the terms "shear stress", "FMD" and "flow-mediated dilatation" associated with the terms "training", "exercise" and "physical activity" in the research to find more work on the topic.

Response: Authors are thankful to the reviewer for this meaning comment. As suggested the additional keywords, ‘flow-mediated dilatation’, ‘FMD’ or ‘shear stress’ were added to the terms "training", "exercise" and "physical activity" in the revised manuscript Lines 73-75. In fact we have used these patterns in our search strategy, and now mentioned the same as suggested.

Comment: In line 94 I suggest explaining the concept of "low quality" so that the reader understands the criterion perfectly, since the term used leaves a vague idea of ​​what it would be.

Response: We are thankful to the Reviewer for this interesting comment. The ‘low quality’ trials mean the papers in non-peer reviewed journals, not indexed in Scopus of Web of Science or preliminary reports. Now this information has been included in the revised manuscript, Lines 94.

Comment: In lines 221 to 225, the authors could present a stratification of ages over 40 years. Between 40 years and 60 years for 20 years. After the fourth decade of life, 10 years generate very important changes, so I suggest that the authors present this data and discuss it in the discussion.

Response: We agree with the Reviewer that the metabolic changes occur in adults after 40 years are important and these changes judge the functioning ability of several organs. In our study, we found resistance exercise effect on BADrest was associated with age. To be specific middle-aged to older adults, means adults with over 40 years were not responsive to RE on BADrest values. The possible reasons for un-responsive BADrest in older adults were explained with suitable references in the revised manuscript, Page 10 and 11, Lines 344-351.  

Comment: In lines 271 to 282 the authors present NO, however, they do not discuss the synthesis and release of this gas. I suggest that you proceed a little further by presenting the synthesis and liberation pathways.

Response: Authors are thankful to the Reviewer for this informative comment. The information about NO synthesis and release has been mentioned in the revised manuscript, Page 9, Lines 271-286.

Comment: Finally, there are other mechanisms that influence endothelial function, such as potassium channels. See the works by Gree et al (2017) and Terwoord et al (2018), attached articles.

Response: Authors are thankful to the Reviewer for suggesting the interesting articles. As directed, the latest evidence of potassium channels in endothelial function has been cited in the revised manuscript, Page 9, Line 286-288.

Reviewer 2 Report

The authors addressed a clinically relevant and timely topic by investigating the effects of resistance exercise on endothelial function. The systematic review and meta analyses are thorough and the results are straightforward, however the English grammar issues make it hard to read and understand at times. I have suggested edits for the first page, after that I stopped correcting grammatical errors but they are prevalent throughout the manuscript. Other than that, the manuscript is of good overall quality and does not need any major revisions.

Abstract:

Line 11: remove ‘intervention’, it is the aerobic exercise that is beneficial, whether it is offered in a research study or just in any clinical situation (where it would not be called intervention).

Line 11: remove ‘the’ before endothelial function

Line 21: add ‘a’ before beneficial effect

Line 22: change to either ‘at rest’ or ‘during resting’

Line 22: change to ‘and age was correlated’ or ‘and the age variable was correlated’

Line 23: ‘represented’ should be ‘presented’

Line 25: change to ‘were not responding’ or ‘were not responsive’ (but not ‘were not responded’)

Line 25: ‘our findings conclude’: this is not possible, findings cannot ‘conclude’ anything on their own. Change into: ‘Based on our findings, we conclude…’

Line 26: instead of ‘better’, use a more precise description (maybe ‘more effective’)

Line 31: remove ‘the’ before CVDs

Line 35: insert ‘the’ before atherosclerotic process

Line 35: change ‘which’ into ‘and’ to make the sentence grammatically correct.

Line 39: insert an s after ‘calculateS’

Line 41: remove ‘the’ before cardiovascular risk

Line 44-45: improving EF through exercise is an important aspect of health promotion does not seem to make a lot of sense as a sentence. Please reword and clarify.

Line 122-123: This sentence seems incomplete and it is not clear what the authors are trying to see about the analysis.

Line 125-130: The method section contains results, please state the approach in the methods and the results in the Results section.

Line 273: Provide appropriate references for the statement that increased shear stress might be a factor why RE improves EF.

Author Response

Responses to Reviewers’ Comments: ijerph-1237253

Reviewer 2 : The authors addressed a clinically relevant and timely topic by investigating the effects of resistance exercise on endothelial function. The systematic review and meta analyses are thorough and the results are straightforward, however the English grammar issues make it hard to read and understand at times. I have suggested edits for the first page, after that I stopped correcting grammatical errors but they are prevalent throughout the manuscript. Other than that, the manuscript is of good overall quality and does not need any major revisions.

Authors Response: We express our sincere thanks to the Reviewer for his/her evaluation and positive comments on our manuscript. We agree with the comments, and we have revised the manuscript accordingly. English grammar mistakes were corrected carefully and the final version of the manuscript was read and corrected by the language expert in the field. All the corrections were marked in red color and detailed response to each comment was given below.

Comment 1: Line 11: remove ‘intervention’, it is the aerobic exercise that is beneficial, whether it is offered in a research study or just in any clinical situation (where it would not be called intervention).

Response: Authors are thankful for the important advice. As suggested, the ‘intervention’ has been removed in the Abstract, Line 11.

Comment 2: Line 11: remove ‘the’ before endothelial function.

Response: The word ‘the’ before endothelial function was removed in the revised manuscript, Line 11.

Comment 3: Line 21: add ‘a’ before beneficial effect.

Response: As suggested, we have added ‘a’ before beneficial effect, Line 21.

Comment 4: Line 22: change to either ‘at rest’ or ‘during resting’

Response: We are apologizing for this typo. As suggested, ‘at resting’ has been corrected as ‘at rest, Line 22.

Comment 5: Line 22: change to ‘and age was correlated’ or ‘and the age variable was correlated’

Response: Authors are thankful to the Reviewer for the keen evaluation of our article. Now we have changed text as ‘and the age variable was’ Line 22.

Comment 6: Line 23: ‘represented’ should be ‘presented’

Response: As suggested ‘represented’ was corrected as ‘presented’ in the Abstract, Line 23.

Comment 7:  Line 25: change to ‘were not responding’ or ‘were not responsive’ (but not ‘were not responded’)

Response: Authors appreciate Reviewer for this suggestion. We have revised the wording accordingly in the Abstract, Line 24.

Comment 8: Line 25: ‘our findings conclude’: this is not possible, findings cannot ‘conclude’ anything on their own. Change into: ‘Based on our findings, we conclude…’

Response: We agree with the Reviewer’s opinion.  As suggested, now we have revised the statement as ‘Based on our findings, we conclude” in the Abstract, Line 25-26.

Comment 9: Line 26: instead of ‘better’, use a more precise description (maybe ‘more effective’)

Response: Authors are thankful to the Reviewer for this meaningful comment. As suggested, the word ‘better’ has been replaced with ‘more effective’ in the Abstract Line 26-27 and also in the Title.

Comment 10: Line 31: remove ‘the’ before CVDs

Response: The word ‘the’ has been removed before CVDs in the revised manuscript, Line 31.

Comment 11: Line 35: insert ‘the’ before atherosclerotic process

Response: As suggested ‘the’ has been added before atherosclerotic process, Line 35.

Comment 12: Line 35: change ‘which’ into ‘and’ to make the sentence grammatically correct.

Response: We are sincerely thankful to the Reviewer for the grammatical comments. As suggested we have changed ‘which’ into ‘and’, Line 35

Comment 13: Line 39: insert an s after ‘calculateS’

Response: As suggested, we have inserted ‘s’ to make the plural form (calculates), in the revised manuscript, Lines 39.

Comment 14: Line 41: remove ‘the’ before cardiovascular risk

Response: The word ‘the’ has been removed before cardiovascular risk, Line 41.

Comment 15: Line 44-45: improving EF through exercise is an important aspect of health promotion does not seem to make a lot of sense as a sentence. Please reword and clarify.

Response: As suggested, the sentences were revised for the clarity. Now the revised sentence (Line 44-45) is giving the clear meaning.

Comment 16: Line 122-123: This sentence seems incomplete and it is not clear what the authors are trying to see about the analysis.

Response: We are apologizing for the unclear sentence. Now we have revised the sentences from Line 120 to 123 for smooth understanding of the meaning.

Comment 17: Line 125-130: The method section contains results, please state the approach in the methods and the results in the Results section.

Response: Authors are thankful to the Reviewer for this comment. We agree with the Reviewer that lines 125-130 contains statistical results. Usually results suppose not to be in the methods section. However, these results are presented to explain the correlations between RE variables and outcomes. Prior to explain the details of subgroups division (based on intensity and age), we wanted to let readers understand the reason why we sub-grouped the trials into low-to-moderate and high-intensity RE and/or young and middle-aged adults. Presentation of meta-regression analysis results in methods sections could unable readers to follow the reasons for such subgroup division so that there will be no gap or confusion to understand the following statements. Therefore, we presented only essential meta-regression analysis results in the methods section. Nevertheless, we revised this paragraph for smooth understanding in the revise manuscript, Page 3, Lines 125-135.

Comment 18: Line 273: Provide appropriate references for the statement that increased shear stress might be a factor why RE improves EF.

Response: As suggested the supporting references for the exercise-induced shear stress and improved EF have been cited in the revised manuscript, Lines 279.
